# A Proof of Principle 2D Spatial Proteome Mapping Analysis Reveals Distinct Regional Differences in the Cardiac Proteome

**DOI:** 10.3390/life14080970

**Published:** 2024-08-01

**Authors:** Wendy E. Heywood, Jon Searle, Richard Collis, Ivan Doykov, Michael Ashworth, Neil Sebire, Andrew Bamber, Mathias Gautel, Simon Eaton, Caroline J. Coats, Perry M. Elliott, Kevin Mills

**Affiliations:** 1UCL Great Ormond Street Institute of Child Health, 30 Guilford Street, London WC1N 1EH, UK; wendy.heywood@ucl.ac.uk (W.E.H.); ivan.doykov@ucl.ac.uk (I.D.);; 2Institute of Cardiovascular Science, University College London, Gower Street, London WC1E 6BT, UK; richard.collis@freeline.life (R.C.); perry.elliott@ucl.ac.uk (P.M.E.); 3Histopathology Department, Great Ormond Street Hospital for Children NHS Foundation Trust, London WC1N 1EH, UKneil.sebire@gosh.nhs.uk (N.S.);; 4Randall Division of Cell and Molecular Biophysics, Muscle Signalling Section, King’s College, London WC2E 2LS, UK; 5Barts Heart Centre, and the Inherited Cardiovascular Diseases Unit, St Bartholomew’s Hospital, West Smithfield, London EC1A 7BE, UK

**Keywords:** heart, proteome, mitochondria, desmoglein-2, proteomics

## Abstract

Proteomics studies often explore phenotypic differences between whole organs and systems. Within the heart, more subtle variation exists. To date, differences in the underlying proteome are only described between whole cardiac chambers. This study, using the bovine heart as a model, investigates inter-regional differences and assesses the feasibility of measuring detailed, cross-tissue variance in the cardiac proteome. Using a bovine heart, we created a two-dimensional section through a plane going through two chambers. This plane was further sectioned into 4 × 4 mm cubes and analysed using label-free proteomics. We identified three distinct proteomes. When mapped to the extracted sections, the proteomes corresponded largely to the outer wall of the right ventricle and secondly to the outer wall of the left ventricle, right atrial appendage, tricuspid and mitral valves, modulator band, and parts of the left atrium. The third separate proteome corresponded to the inner walls of the left and right ventricles, septum, and left atrial appendage. Differential protein abundancies indicated differences in energy metabolism between regions. Data analyses of the mitochondrial proteins revealed a variable pattern of abundances of complexes I–V between the proteomes, indicating differences in the bioenergetics of the different cardiac sub-proteomes. Mapping of disease-associated proteins interestingly showed desmoglein-2, for which defects in this protein are known to cause Arrhythmogenic Right Ventricular Dysplasia/Cardiomyopathy, which was present predominantly in the outer wall of the left ventricle. This study highlights that organs can have variable proteomes that do not necessarily correspond to anatomical features.

## 1. Introduction

The sequencing of the human genome provided a significant leap in how we understand the human body and disease. Today, this information is being used more and more in precision medicine and diagnostics. However, there are still fundamental questions that cannot be answered by genetics alone. To demonstrate this point, we set out to proteomically map a heart to determine if there are significant and subtle changes in the proteins of the heart across a dissected 2D spatial plane. Unlike the genome, the proteome is highly variable across an organism and dynamic over time [1]. It is influenced by spatial patterning, prior cellular differentiation, biochemical requirements, and effects of the local or systemic environment. The proteome, therefore, provides a better explanation of the role or requirements and ‘instantaneous phenotype’ of a cell or group of cells [2]. Subsequent studies of mouse and human brain tissue have widened this concept to also describe variation in the regional ‘lipidome’ [3,4] and ‘metabolome’ [5]. Such variability is suggested, by multiple lines of evidence, to occur both between whole organs and across individual tissues. 

Organ mapping studies of small molecules such as lipids and metabolites are traditionally performed using mass spectrometry imaging techniques, where a 2D pane can be fixed to a surface and subjected to either a laser (MALDI) or electrospray (DESI) ionization method. These techniques work well for small molecules. Proteins, however, are more challenging as protease digestion to correctly identify unique signature peptides is required. MALDI IMS is emerging as the superior method for tissue imaging of proteins [6]; however, it still does not provide the identification of proteins and, hence, no information on biochemical function or requirements of that area of the heart. Tissue imaging of the heart is also limited due to the large insoluble and highly crosslinked nature of cardiac tissue proteins. Advances have been made using de-cellularisation techniques to look at the extracellular matrix of the heart proteome; however, surface-based tissue imaging methods are still limited in the number of proteins that can be detected when compared to conventional liquid chromatography methods [7]. 

Although the heart functions in a highly coordinated manner, there is considerable regional variation in its phenotype and function. Several distinct divisions are easily recognised, including the atria and ventricles, the left and right heart, the epicardium and endocardium, and the contractile and conductive tissues. Beyond this, individual cardiomyocytes continuously adapt to the unique demands placed upon each of them. Underlying this phenotypic variation is a complex and diverse map of differentially produced and modified proteins. Many cardiac diseases manifest themselves within distinct regions of the heart. Though often poorly understood, this phenomenon may be explained by underlying differences in the local transcriptome and proteome. Common pathologies, such as heart failure or myocardial ischaemia, can affect any cardiac area. Both, however, have recently been shown to cause discrete changes in the local proteome, depending upon which ventricle they affect. In response to pressure-overload heart failure, Friehs et al. [8] demonstrated that the right ventricle preferentially up-regulates contractile and calcium-handling proteins, whilst the left ventricle increases oxidative phosphorylation capacity.

Various proteomic studies have compared the cardiac proteome in animal models and humans by anatomical region [9,10,11,12]. However, we set out to fully proteomically characterise a whole section of heart to see if (i) the degree of proteome changes with very different anatomical and physiological function and (ii) which biochemical pathways are more prevalent or important in different areas and whether this could be explained by the physiological requirement of that area of the heart. Figure 1 shows the overall experimental design of the study, where we used careful tissue sectioning and analyzed each section using label-free proteomics in order to detect enough proteins to get a sense of the proteomic variation and whether it is relevant to the anatomical region or not. This ‘eyes open’ approach has identified unknown regions of significant proteomic variation. We have also looked at the proteome with respect to key biological functions of the heart, such as the distribution of sarcomere proteins and mitochondrial metabolism and its relation to the anti-oxidome.

## 2. Methods

### 2.1. Heart Tissue Sectioning

The bovine heart model was chosen for this study as it has often been used for cardiology studies, and the bovine reference proteome is also better annotated than other species’ proteomes. A complete bovine (*Bos taurus*) heart was obtained within 12 h post-mortem.

Specific ethical consent was not required since the animal was sacrificed for other purposes. Total heart mass was 2110 g. It was maintained on ice and protected from direct contact to avoid freeze-thaw. Assuming a ‘valentine-heart’ orientation, a coronal, four-chamber slice was prepared. The heart was photographed, and subsequently, sections relating to Figure 2A were excised using a No 20 curved scalpel blade. This number of samples was considered to offer good coverage of all major cardiac regions. Each ventricular free-wall and the septum were divided into three longitudinal slices. Within the free-walls, these represented the endocardium, mid-myocardium, and endocardium. Each of these was sliced perpendicularly into ten rows, creating a total of 90 ventricular sections. The papillary muscles were similarly divided into 4 to 5 ‘short-axis’ sections. The vestibules of each atrioventricular valve were removed and divided radially into five sections. Similarly, five sections were taken along the length of the lateralmost angle of each atrial appendage. Finally, two and three sections were isolated from the moderator band and the pulmonary infundibulum, respectively. From each section, 4 mm × 4 mm × 4 mm cubes were taken as the final sample after removing any fatty or connective tissue. The samples were taken from a representative part of each section; from an endocardial section, for example, the sample was obtained from the endocardial surface. The four-chamber slice was maintained on ice throughout this process. All sections were washed with cold water until all visual evidence of blood contamination was removed, then immediately placed into 1.5ml microcentrifuge tubes and stored at −80 °C.

### 2.2. Tissue Preparation

Frozen samples were powdered by hand using a pestle and mortar. To maintain consistency, thirty strokes were performed upon each before immediate return to the freezer. A total of 1 mL of lysis buffer (50 mM Ammonium Bicarbonate; 2% *w/v* Amidosulfobetaine-14 (ASB-14)) was added and left to shake for 1 h. Each sample was homogenised using a handheld glass homogeniser. Homogenised samples were finally subjected to probe sonication to further disrupt cellular structure. This was set to a 10-micron amplitude and applied three times for ten seconds. Samples were maintained on ice throughout as much of the procedure as possible. Homogenates were centrifuged at 13,000× *g* for 10 min to remove insoluble components.

A total of 100 μL of supernatant was diluted 10:1 with lysis buffer and assayed for protein content using a bicconinc acid assay kit (Thermo Scientific, Waltham, MA, USA); 150 μg of each sample was freeze-dried.

### 2.3. Sample Digestion

Samples were reconstituted in 20 μL of digest buffer solution (100 mM Tris-HCl; 2% *w/v* ASB-14; 6 M urea; 2 M thiourea) and double digested, as previously described [13]. Briefly, samples were incubated in 3 μL of 100 mM Tris-HCl pH 7.8 containing 20 mM dithioerythritol and incubated at room temperature for 60 min. Free thiol groups were carboamidomethylated by incubation with 6 μL of 100 mM Tris-HCl, pH 7.8, containing 20 mM iodoacetamide, and incubated at room temperature for 45 min. The reaction mixture was then diluted with 155 μL of H_2_O and vortexed, and 150 ng of Lys-C trypsin was added to the solution. Samples were incubated overnight (12–16 h) at 37 °C in a water bath. Digests were subsequently desalted using C18 columns (Isolute, Jones Chromatography, Hengoed, UK) and dried by centrifugal evaporation. 

### 2.4. Label-Free Proteomics

Digested samples were analysed by shotgun UDMSe [14], as described previously [15]. All samples were run in one batch with a peptide mix used as a QC every 15 injections. Briefly, peptides were separated by reverse phase chromatography over a 60 min gradient on a Waters Nanoaquity system coupled to a Waters Synapt G2 Si mass spectrometer. Data were acquired by MSe DDA with ion mobility. Progenesis QI analysis software (version 3.0) was used to identify peptides against an initially uniprot bovine reference proteome, and then a second pass search was run against a human reference proteome. Fixed modifications included carboamidomethylated cysteines and variable oxidised methionine. Search criteria included a minimum of 1 missed cleavage, 1% FDR, a minimum of 5 fragment ions per peptide, 7 per protein, and 2 unique peptides per protein. Normalised abundance data were exported into Excel for subsequent analysis. The mass spectrometry proteomics data have been deposited to the ProteomeXchange Consortium via the PRIDE [16] partner repository with the dataset identifier PXD013375 and 10.6019/PXD013375.

### 2.5. Data Analysis

Multivariate analysis was performed using SIMCA version 14 (Umetrics, Umeaa, Sweden) on normalised exported data from Progenesis QI analysis software. Proportion and multiple correlation analysis was conducted using Microsoft Excel. Heatmaps were generated by using http://www.heatmapper.ca/ (accessed on 24 July 2024) [17]. Graphpad prism v6 was used for statistical analysis of complexomics data and volcano plots. Gene ontology analysis was performed using the Panther GO functional enrichment and overrepresentation analysis. Comparative analyses to identify cluster-enriched proteins for heat maps and volcano plots were performed using Progenesis, and ANOVA p values, fold change, and normalised data were exported to Excel. A 2D image mapping tool was developed using Visual Basic developer for Excel 2013 (Microsoft, Redmond, WA, USA), along with PaintShop Pro X6 v.16.0.0.113 (Corel, Ottawa, ON, Canada) for image editing. 

### 2.6. Data Visualisation

Sectioned heart map images showing intensity heat maps of selected protein or complex value abundance across the heart were created using Visual Basic developer for Excel 2013 (Microsoft, Redmond, WA, USA), along with PaintShop Pro X6 v.16.0.0.113 (Corel, Ottawa, ON, Canada) for image editing. Interactive heat maps for clinical proteins of interest and for some selected proteins from the PCA analysis are included as Appendix A.

## 3. Results

### 3.1. Multivariate Analysis of the Bovine Heart Proteome

The bovine heart was carefully tissue sectioned, as depicted in Figure 2, where a four-chamber slice was cut into over 130 cubes sized 4 mm × 4 mm × 4 mm. The sections were individually analysed using an unbiased proteomic approach. A total of over 2000 proteins were identified in the experiment. All protein abundance data were analysed by principal components analysis (PCA). Figure 3A shows that the heart proteome was found to cluster into three distinct proteomes. A loading plot (Figure 3B) derived and corresponding to the PCA plot in 3a depicts the proteins that drive the actual difference in these clusters. Figure 3C shows the regions identified by proteome differences in Figure 3A mapped to the heart organ to see how they relate to the anatomical regions. The region with the greatest variation (termed cluster 1) was the outer layer of the right ventricle (RV) wall, as depicted in the heart map in Figure 3C. A second cluster (termed cluster 2) comprises sections from the outer-basal portions of the left ventricular free wall, the tricuspid and mitral valves (AV ring), and the right atrial appendage. The third cluster consists largely of intraventricular septum, apical and inner basal portions of the left and right ventricular free walls, and the left atrial appendages. Thus, these areas seem to have similar proteomes, which we termed clusters 1, 2, and 3 in this study. Within the clusters themselves, there is further sub-level variation, as shown in Figure 3A, where the left atrial appendage (LAP) sections almost cluster independently from cluster 3. Further analysis of cluster 3 to see if any of the other region sub-clusters (apart from LAP) did not reveal further sub-level variation for this proteome/myocardial tissue (Figure 3D). However, further analysis of cluster 2 (Figure 3E) demonstrated that the anatomical regions that make up this cluster separate from the outer left ventricular wall having its own sub-proteome. Both the mitral and tricuspid valves also have different proteomes compared to the other regions whilst also having significant but subtle differences in their proteomes relative to one another. The observation that the proteomes of the outer RV wall and, to a lesser degree, the outer left ventricular (LV) wall are very distinct is unexpected. Technical reasons for this observation were ruled out as biopsies were randomised during preparation and analyses. There are reports showing that the morphology of the RV is distinct from the rest of the heart [18,19,20]. We attempted to look at cardiac cell type distribution based on previously described cardiac cell types [21] to see if this could account for the RV outer wall change. We were able to detect protein markers of cardiomyocytes (troponin T), smooth muscle (transgelin), fibroblasts (vimentin), and adipocytes (adiponectin) (Appendix A). Troponin T was uniformly abundant throughout the heart apart from a strong enrichment in cluster 1 RV outer wall. Smooth muscle is low across the heart but enriched in the left atrial appendages. Fibroblasts largely follow the profile pattern of cluster 3 in the inner region of the heart with distinct low abundance in the RV outer wall and back LV wall. Adipocytes have a low-level expression but are slightly enriched in the back LV wall and valves.

### 3.2. Protein Composition of the Heart Proteomes

To look closer at what drives the differences between the regional proteomes or clusters, as identified by multivariate analysis, we first looked at the composition of the most abundant proteins in each of the clusters. Sections assigned to clusters were grouped. Average values for each protein were calculated for the clusters. These values were used to calculate the percentage value of each protein for each cluster. These percentages were ranked, and the most abundant proteins that contributed to more than 1% of the cluster proteome are shown in Figure 4. 

The dynamic range of protein abundance is depicted by the ranked abundance graph accompanying each cluster pie chart in Figure 4. This indicates the proteomes from each cluster are overall quite heterogeneous and not dominated by common proteins, unlike other biological proteomes such as serum. Cluster 1 (outer RV wall) shows 15 proteins that constitute >1% of the cluster 1 proteome. Together, they comprise approximately 27% of the total proteome for cluster 1. The most abundant proteins are myosin 7 (3.1%) and protein-l-isoaspartate (d-aspartate) *O*-methyltransferase, also known as PIMT. PIMT is not abundantly expressed in the other proteomes. PIMT recognises and catalyses the repair of damaged L-isoaspartyl and D-aspartyl groups from spontaneous deamidation in ageing proteins and prevents cardiomyocytes from hypoxia-induced apoptosis [22]. The abundance of this protein indicates that this region may be subjected to potential levels of stress and, therefore, require this mechanism of repair. Proteins that were observed to be consistently abundant in every proteome included serum albumin, myoglobin, and the mitochondrial-associated proteins ATP synthase subunit beta, mitochondrial uncoupling proteins 4, and creatine kinase. One of the most abundant proteins for cluster 3, outer LV wall, and LAP included HYDIN ‘Axonemal Central Pair Apparatus Protein’, a cilia-associated protein, which is associated with primary cilia-associated dyskinesia [23]. Cilia are known to play a role in congenital heart disease pathogenesis as cilia are required for heart development [24]. Recent whole exome sequencing has revealed variants in HYDIN are associated with sporadic atrial septal defects [25]. In the adult human heart, cilia are not abundant.

A heat map of the significant and differentially abundant proteins is shown in Figure 5A, with volcano plots used to demonstrate and compare each proteome with the rest of the other proteomes (Figure 5B). Statistical analysis comparing the sections according to the cluster indicated by the original PCA analysis showed that 445 proteins were significantly differentially abundant in cluster 1. A total of 259 of these proteins were observed to be specifically enriched compared to the rest of the cardiac proteome. Cluster 2 had 747 significantly differentially abundant proteins, of which 238 were enriched. Cluster 3 has 750 significantly altered proteins, of which 483 were more abundant in cluster 3. The sub-cluster 2A, consisting of the outer left ventricle wall, demonstrated 181 proteins significantly altered compared to all other clusters, including the rest of cluster 2. The LAP section showed 151 enriched proteins from the rest of the cardiac proteome.

Gene ontology analysis of the enriched proteins indicated that most of the differences in biological function were observed to be involved in cellular energy metabolism. Cluster 1 did not show any specific functions for this region. However, cluster 2 had specific functions for cardiac muscle-related functions, such as contraction and development. Furthermore, proteins enriched in Cluster 3 were known to be involved in fatty acid metabolism, immune response, and detoxification pathways (Appendix A). One finding of interest is the observation of the enrichment of nebulin in cluster 3 (Figure 5B). The role of nebulin in the heart is little understood and is thought to be produced predominantly in skeletal muscle and in low abundance in cardiac muscle [26,27]. Using mass spectral analysis, nebulin is observed to be in the top 75% of the cardiac proteome by Doll et al. [12], and our study is able to detect nebulin as differentially produced across the heart. This altered tissue distribution could account for some of the contention surrounding the detection of nebulin. 

### 3.3. Myosin/Sarcomere Protein Distribution in the Heart

Muscle contraction is a key biological function in the heart, with myosins being one of the most important proteins in this process. Defects in various myosins and associated proteins can result in cardiac disease. Therefore, we decided to look at the sarcomere-associated protein distribution across the heart proteomes. Figure 6 shows a clustered heat map distribution of the sarcomere-associated proteins, and Appendix A shows cardiac M-band protein analysis. This heat map shows the distribution of different myosins is similar across the heart. The LAP region has very high levels of myosin heavy chains 2 and 13 and myosin light chains 4, 6B, and 7. MYL4 and MYL7 (also known as atrial light chain-2) are known to be specifically produced in adult atria, which our findings confirm [28]. Cluster 1 shows enrichment of heavy chains 1, 6, 7, 7B, and 8, MYBPC, and cardiac troponin. MYH8 is thought to be a fetal myosin; however, we see a low level of abundance in adult bovine hearts. Whether this observation stands for human heart tissue remains to be determined. MYH1, also known as striated muscle myosin heavy chain 1, is known to be highly abundant in fast type IIX/D muscle fibres and smooth muscle cells [29], indicating specific enrichment of this cell type in cluster 1. MYH6 is abundant in both ventricular and atrial tissue but is enriched more in the atria [30]. Our data indicate it is also more abundant in the outer wall of the RV. MYH7 is the most abundant protein in cluster 1 and is the major protein comprising the thick filament in cardiac muscle. Cluster 3 has higher levels of myosin heavy chain 16, actin c1 (cardiac actin), and light chain 3, which is known to be a ventricular myosin [31]. Cluster 2, including the outer wall, has enriched levels of the unconventional myosins 5A, 9A, and 9B. Unconventional myosins are largely involved with the transport of organelles along actin fibres in the cell but do have other functions, such as actin organisation, mitotic spindle regulation, and gene transcription [32]. We have also included ankyrin repeat domain proteins and myozenin, which localises with the cluster 1 proteins. These overall observations indicate that the outer RV wall appears more muscular than the other regions of the heart. The RV wall is known to be thinner than the LV wall, which contradicts these findings. One possible reason for this is that muscle tissue dissociation in the homogenisation process may be improved by the reduced thickness/compactness, allowing for better protein solubilisation and digestion. 

### 3.4. Mitochondrial Protein Analysis

Many of the differences observed between clusters were proteins and pathways involved in energy metabolism and, in particular, were mitochondrial in origin. Therefore, we decided to look, in more detail, at the mitochondrial-associated proteins. Due to the abundance of mitochondria found in each section, we were able to successfully develop, stratify, and interrogate the ‘mitochondriome’ in each individual cluster. Mitochondrial proteins were extracted from the data set by using the PATHER GO functional classification annotation tool. Proteins were further sorted into mitochondrial complex proteins according to HGNC (genenames.org).

The data set contained 30/44 complex I (all supernumerary subunits), 2/4 complex II subunits, 7/10 complex III subunits, 6/19 complex IV subunits, and 12/20 complex V subunits. Those subunits not detected reliably was due to them being present at the lower limits of detection. The total subunit values were standardised by ratio to citrate synthase, a standard marker of mitochondrial abundance and, hence, a marker of mitochondrial distribution. We confirmed this reflects mitochondrial abundance by confirming the correlation of citrate synthase levels with other known mitochondrial proteins Hydroxyacyl-Coenzyme A dehydrogenase, voltage-dependent anion-selective channel protein 3 and 2, and aconitase 2 (Appendix A). The abundance of citrate synthase is shown in Figure 7A and is variable itself across the heart mitochondria, being particularly enriched in cluster 3 in the inner walls of the heart and the septum. When standardising to citrate synthase, there are differences in the abundance of the individual complexes between proteomes (Figure 7B). Complex I alone is significantly increased in cluster 3. Complexes II–V are more abundant in cluster 2 and the outer LV wall compared to clusters 3 and 1. Complex IV is more abundant in cluster I. The differences in the relative abundance of the complexes within the clusters are shown in Figure 7C, which re-iterates that complex IV is specifically upregulated in cluster 1, and complex I is the second most abundant complex in cluster 3. The differential abundance of mitochondrial complexes is not completely understood but may, in part, relate to the variable formation of super complexes and more densely packed mitochondria. The different ratios of the individual complexes may reflect the energy production (glucose vs. lipid) of the cells in that area of the heart. However, it has been hypothesised that the respiratory enzymes of the mitochondria are organised into super complexes. This increases the metabolism efficiency and reduces oxidative stress damage. Different levels of complex I in super complexes have been observed between brain cell types [33] and determine differential ROS production.

Mitochondrial function is a major function in the heart and has been studied extensively. Defects in mitochondrial metabolism tend to affect the two energy-demanding organs that require it the most—the brain and the heart. Therefore, it is plausible that super complex changes occur in the heart in different regions or cardiac cell types as they appear to do in the brain. The greater abundance of complex IV in the outer RV wall, along with the previous observation of enrichment of the stress response protein PIMT, does point to the fact the outer RV wall undergoes significant stress and demand for respiratory capacity. A previous study has found that myoglobin abundance can modulate complex IV abundance [34], but we were unable to observe a relationship between myoglobin and Complex IV levels in this data set. To try to understand why there may be alterations in the abundance of the mitochondrial complexes across the heart, we also looked at the abundance of key proteins involved with fatty acid oxidation to see if this changes. We observed that cluster 2a in the outer LV wall had consistently abundant lower levels of these FA oxidation enzymes compared to the rest of the heart. This implies that this region utilises glucose for energy, more so than fatty acids. (Appendix A). 

Our observation that there were subtle differences in the relative amounts of the mitochondrial complexes, which are known to be responsible for producing more free radicals than others, led us to determine if these changes could result in the heart reflecting this phenomenon by correspondingly upregulating those proteins with antioxidant properties in these clusters. In particular, with the observation of increased abundance of complex I in cluster 3, we examined the distribution of antioxidant proteins as mitochondrial ROS production, which is thought to play an important mechanism of disease in the cardiovascular system [35]. We compared clusters 2 and 3 as there were insufficient data in cluster 1 for this analysis (Figure 8). 

Using a multiple correlation matrix, we observed a relationship between the mitochondrial complexes themselves within cluster 3. SOD2—the main antioxidant involved in mitochondrial-derived RO production—has an association with most of the complexes, whilst cytosolic SOD1 has hardly any association with the abundance of the complexes. The other system that regulates ROS is glutathionylation by the glutathione S transferases (GSTs). Not surprisingly, there is a strong correlation in cluster 3 with both of the detected GSTs, Glutathione-*S*-transferases P and mu, which are mitochondrial and cytosolic GSTs, respectively [36]. GPX1, an important enzyme in the mitochondria, is involved in the reduction of the potential free radical-producing hydrogen peroxide and appears to have an association with an abundance of complexes IV and V in cluster 2. Out of the peroxidoxins, PRDX 2 and 6 appear to have a stronger association with the complexes in cluster 3 than 2. The mitochondrial PRDX3 has an association in both clusters 2 and 3 but not with complexes I and II in cluster 3. PRDX6 has a much stronger association with the complexes in cluster 3. PRDX6 also acts to reduce not only hydrogen peroxide but also short-chain organic fatty acid and phospholipid hydroperoxidase; however, its role in the heart is not well understood, but reduced abundance is thought to make the heart vulnerable to ischemic injury [37]. As the heart preferentially utilises fatty acids for energy over glucose, it is unsurprising that this PRDX is associated with mitochondrial metabolism. An interesting finding is that this occurs in the cluster 3 region, which comprises the myocardium and inner walls of the heart. Complex I is the main complex that generates mitochondrial [35] ROS and is elevated in cluster 3. It is, therefore, unsurprising that cluster 3 has a greater association with antioxidant systems, as shown in Figure 8. This indicates that this region of the heart is more metabolically active and, thus, requires commensurate tighter control of the redox system. 

### 3.5. Distribution of Disease-Associated Proteins across the Heart 

Using the proteomic data set, we selectively chose those proteins that are known to be associated with familial forms of hypertrophic (HCM), dilated (DCM), and arrhythmogenic right ventricular cardiomyopathy. The abundance data for twenty-eight proteins were extracted from the data set heat-mapped (Figure 9A) and clustered to anatomical regions to identify if there was tissue-specific distribution. Many of the disease-associated proteins had low abundance in the atrium, papillary muscles moderator band, and tricuspid and mitral valves. Many of the proteins overlap in relation to cardiomyopathy, with various mutations resulting in either HCM or DCM. However, there did not appear to be an HCM/DCM-predominant associated abundance of the proteins according to anatomical region. A higher abundance for 11 proteins indicated on the heatmap as box 1 was observed in the intraventricular septum and LV. To illustrate this, the abundance of one of the proteins, troponin C, was mapped to the 2D image of the heart in Figure 9B. Another grouping of high protein abundance was observed in the outer wall of the right ventricle, termed box 2. One of these proteins, myosin heavy chain 7, is heat mapped to the 2D heart image in Figure 9B to illustrate this. Interestingly, the heart map also shows a strong abundance of desmoglein (DSG2) in the outer wall of the left ventricle (Figure 9B). DSG2 is a specific cadherin of the cell–cell contact in cardiac desmosome and is involved in the pathogenesis of arrhythmogenic cardiomyopathy [38], which often affects the right ventricle. There have been reports that DSG2 mutations are associated with a more LV-predominant phenotype than most other gene mutations A(R)VC mutations [39]. This finding that the abundance of DSG2 is higher in the outer LV wall may explain the involvement of the LV in DSG2 ARVC. In HCM troponin I and MYBPC3, mutations have been associated in some cases with more asymmetric septal hypertrophy [40,41], which corresponds to the finding in our study that the abundance of these proteins is strong in the intraventricular septum.

## 4. Discussion and Conclusions

Recent work has used single-cell transcriptomics analysis to characterise cardiac cells types [21,42], which show cellular heterogeneity of cardiac cell types that have distinct atrial and ventricular profiles. However, the transcriptome does not directly relate to the proteome, especially in tissues [43]. Single-cell proteomics in cardiology is in its infancy but is coming [44], and combining it with a spatial 2D map approach will be a very powerful technique. 

The heart is composed of various cell types, of which cardiomyocytes consist 30–40% of the cell population (70–85% of the heart volume); the remaining population consists of fibroblasts, endothelial cells, and smooth muscle cells [26]. Our analysis indicates there is a cell type distribution relating to the clusters, with cardiomyocytes highly enriched in cluster 1 in the outer RV wall and fibroblasts dominant in cluster 3 in the inner regions. Unfortunately, we were not able to determine an endothelial proteome signature. 

The unbiased extensive mapping approach of this study has involved the excision of >130 sections in order to cover the proteome down to the sub-centimetre/millimetre order. With label-free proteomics being performed on each section and totalling over 150 h of analysis time, the scale of this analysis has meant that only one heart plane has been practicably possible to dissect and analyse, and single shotgun UDMS^e^ runs were performed per biopsy, thus limiting detection of lower abundant proteins. This very exciting pilot project demonstrated the potential for performing spatial proteomics of the heart; however, due to extensive mapping of only a single heart, we cannot account for biological variation, which could be affected by age, sex, and breed. This work confirms the potential for spatial heart omics and should be extended to performing more analyses on more hearts in relation to disease and age. As a single heart has been used, the abundance analysis is normalised across the heart, and whilst direct abundance values are related to that specific heart, we expect the normalised changes or profile observed to be relatable to other hearts if profiled extensively in this way. 

Due to this limitation, this work stands as a proof of concept study showing proteomic changes are not always defined to anatomical regions. By taking an in-depth look at the actual distribution of protein abundance across a complete four-chamber 2D plane of the heart, we have been able to observe distinct proteomes that change from the inner to outer, from apex to base, and across multiple chamber walls, including important structures of the heart that targeted anatomical sampling would miss. 

This concept of non-anatomical-driven proteome changes likely extends to other species, such as humans, and this work provides a foundation for this analysis. This 2D spatial profiling could be the basis of a larger study to confirm if human hearts also have non-anatomically defined proteomes. Limitations can be overcome if a small number can be in-depth profiled, and as cells are defined by markers, the regions could be defined by a profile of proteins, which could be used for targeted high throughput analysis on larger sample numbers. This approach, combined with the power of single-cell proteomics of newly discovered non-anatomical regions, could really provide a deep understanding of the human heart. 

Significance statement: This study highlights that organs can have variable sub-proteomes that do not necessarily correspond to anatomical features. The bovine heart, as a model, demonstrates at least 3–5 separate proteomes exist within a 2D plane of the heart. The main differences between these proteomes were the biochemical pathways utilised energy substrates. By mapping individual protein abundance across a 2D image, we can see where proteins of interest, such as disease-associated proteins, are specifically produced and how specific areas of the heart have subtle but significant differences in the ratio of mitochondrial complexes. This may provide an idea of where to focus analysis on in the cardiac proteome for future studies and into the different mechanisms associated with function, stresses, and potential disease processes.

## Figures and Tables

**Figure 1 life-14-00970-f001:**
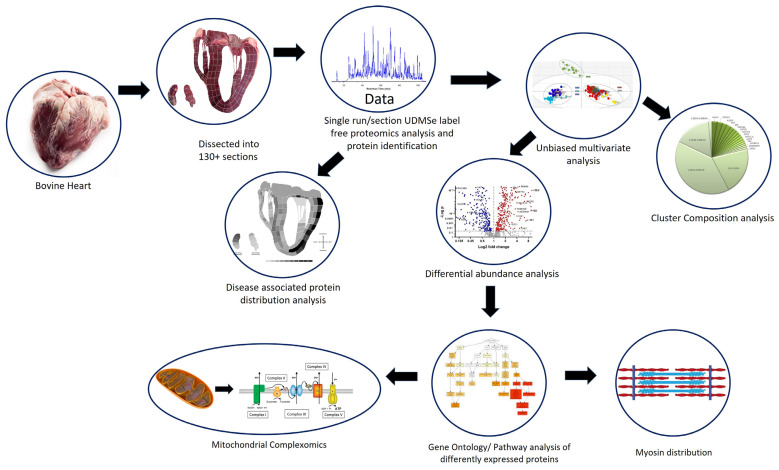
Overview of the experimental design. Source material consisted of a single bovine heart cut into >130 sections. Each section was homogenised and analysed using shotgun label-free proteomics. In total, >1000 proteins were confidently identified. Bioinformatics analysis involved mapping the abundance of known cardiac disease proteins. Unbiased multivariate analysis was used to identify regions defined by their proteome. These regions/clusters were analysed for protein composition and differential protein analysis. Differentially abundant proteins were subjected to pathway analysis, which indicated many altered functions were associated with structural protein distribution and mitochondrial function, which led to sub-analysis of the mitochondrial complexes.

**Figure 2 life-14-00970-f002:**
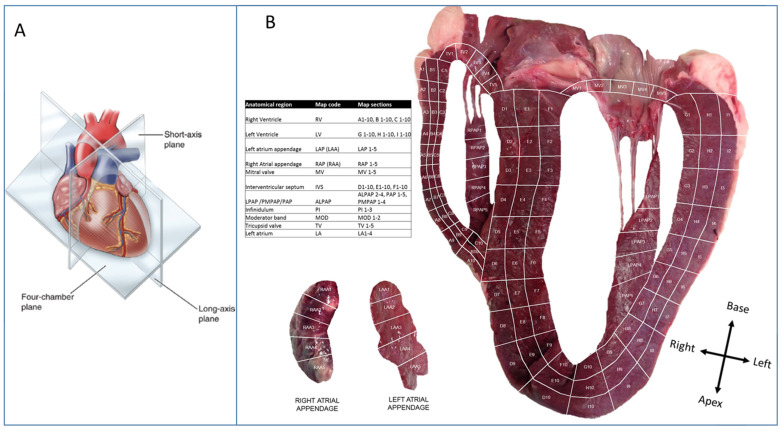
Section plan for heart map image. (**Panel A**): Schematic diagram illustrating how the heart was cut according to the four-chamber plane. (**Panel B**): The bovine heart cut, as illustrated in panel B, is sectioned as indicated in the image. Each section relates to an anatomical region that is listed in the accompanying table.

**Figure 3 life-14-00970-f003:**
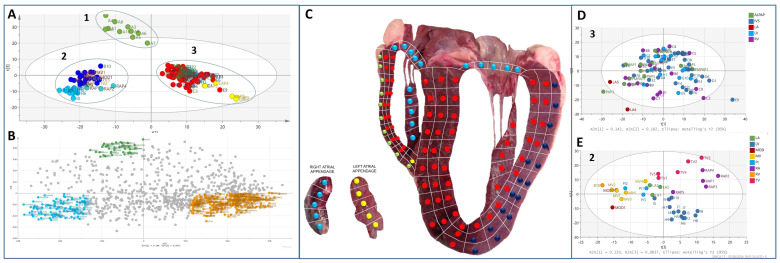
Principle component analysis (PCA) of the 131 heart regions based on their proteomic abundance profiles. (**A**) shows the proteomes of all sections distinctly cluster into three main groups. The first and third components segregate the heart sections and account for 19.8% and 8.7% of the variability, respectively. (**B**) shows the PCA analysis loading plot, showing proteins driving the separation between the three clusters. (**C**). Proteome map. The clustered regions depicted in A are colour-coded and mapped on the 2D image of the heart, illustrating the spatial profiling of the different proteomes (**D**) depicts the PCA analysis of cluster 3 proteins (excluding LAP sections), showing there is no sub-clustering of these sections, and cluster 3 is heterogeneous. (**E**). depicts PCA analysis of cluster 2 from A, which shows further segregation of specific anatomical regions with similar proteomes.

**Figure 4 life-14-00970-f004:**
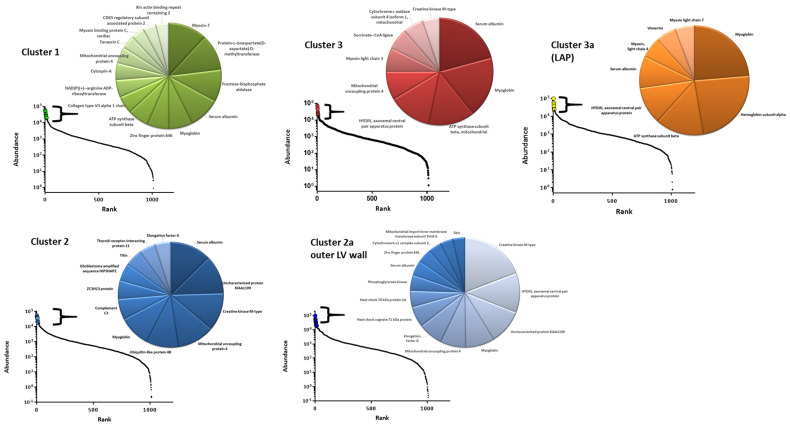
Protein abundance and dynamic range of protein abundances of the clusters and regions identified by PCA analysis. PCA clusters 1–3 are depicted. The sub-clusters of cluster 2 (LV outer wall) and 3 (LAP) are treated separately. Graphs show dynamic range by abundance versus rank. The most abundant proteins that consist of >1% of each cluster/region proteome are shown in the corresponding pie chart. Clusters 1 and 2 (including the LV outer wall) show the most heterogeneity in abundant proteins, whilst cluster 2 and LAP are dominated by myoglobin, albumin, and haemoglobin, proteins driving differences in the individual clusters of the heart proteome.

**Figure 5 life-14-00970-f005:**
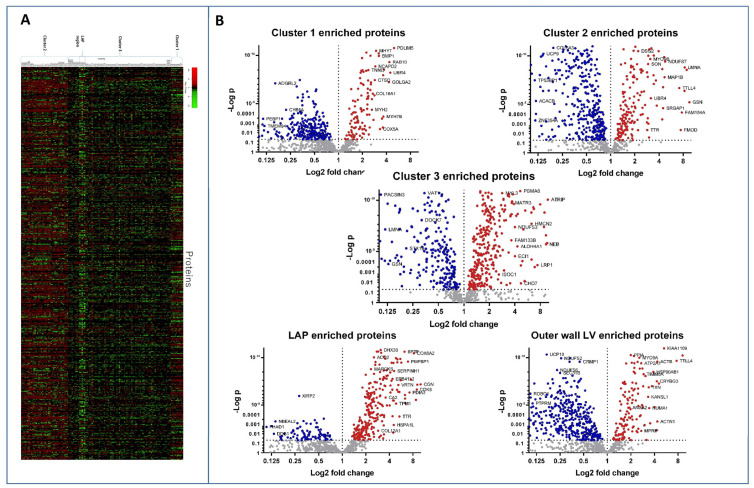
Proteins significantly differentially produced across the heart proteome. (**A**) Heat map of proteins significantly differentially produced across the bovine heart. (**B**) Volcano plots of each region compared to all other regions identify proteins specifically enriched in each cluster.

**Figure 6 life-14-00970-f006:**
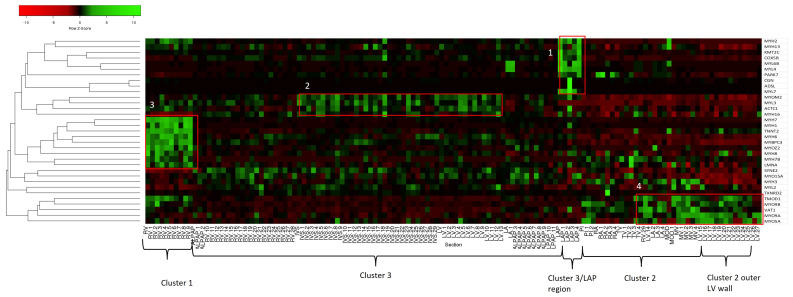
Contractility Heat Map. Cluster analysis of standardised abundance of sarcomere-associated proteins. The highlighted regions indicate enrichment of specific contractile proteins to different anatomical regions. Box 1 indicates enriched contractile proteins in cluster 3. Box 2 indicates atrial-specific myosin heavy chains 2 and 13 and myosin light chains 4, 6B, and 7 cluster with various ankyrin repeat proteins. Box 3 highlights proteins enriched in cluster 1 and the RV. Box 4 indicates unconventional myosins are predominantly produced in cluster 2 regions.

**Figure 7 life-14-00970-f007:**
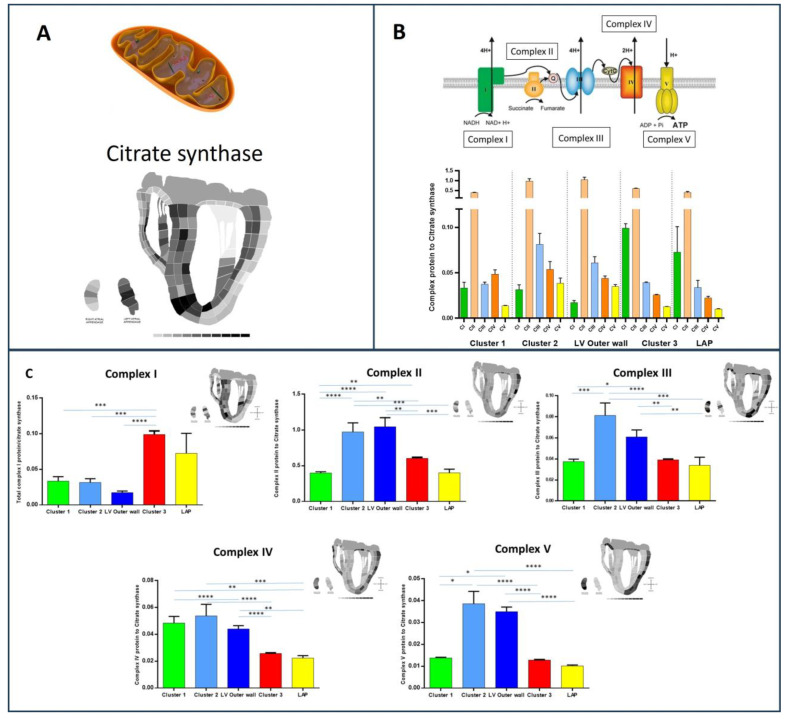
Complexomics of the bovine heart. (**A**) shows the total levels of citrate synthase across the heart, which indicates enrichment of citrate synthase and, hence, mitochondria in cluster 3 in the inner myocardium of the heart. (**B**) shows values of all complexes ratioed to citrate synthase together for each cluster to observe the pattern of abundance. The pattern of abundance of the complexes appears different in cluster I as there are higher levels of complex IV than III. As depicted in B, cluster 3 shows much higher levels of complex I. (**C**) shows the total detected complex protein ratio to citrate synthase for each proteome region identified in Figure 2. The abundance of each complex is depicted on the 2D map heart for each complex. There are higher levels of complex I in cluster 3, whilst there are enriched levels of complexes II–V in cluster 2 and the LV outer wall. * represents *p* < 0.05, ** represents *p* < 0.01, *** represents *p* < 0.001, **** represents *p* < 0.00001.

**Figure 8 life-14-00970-f008:**
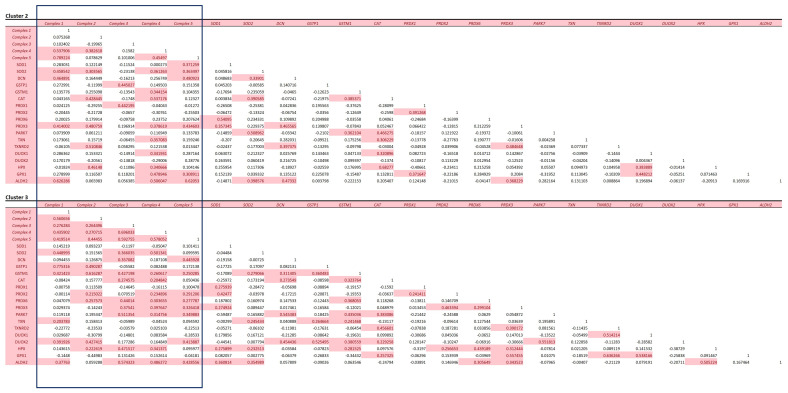
Mitochondrial complex association with antioxidant proteins. Spearman correlation matrices for total protein mitochondrial complexes I–V and known antioxidant proteins for clusters 2 and 3. Highlighted cells are considered to have significant correlation, and the values on the cell are r^2^ for how well the proteins correlate with individual complexes or other antioxidant proteins. Cluster 3 has more associations with antioxidant proteins with a particular strong association between complex I and GSTP1. The peroxidoxins appear more associated with the latter complexes 3–5.

**Figure 9 life-14-00970-f009:**
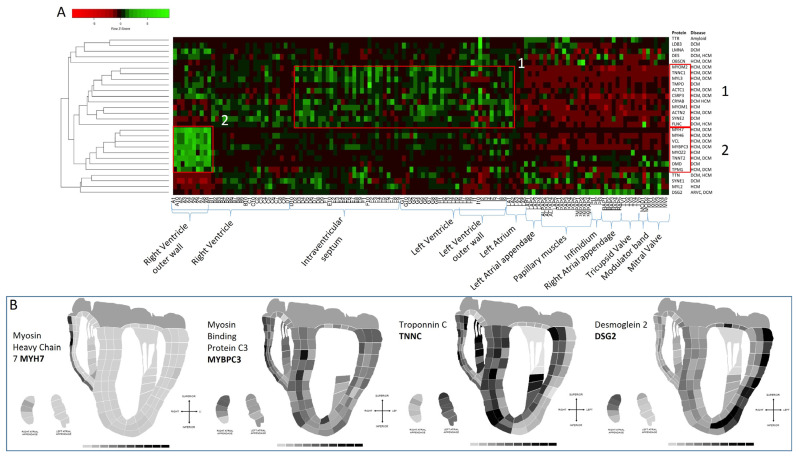
Abundance of known disease-associated proteins across the heart. (**A**) Heat map analysis of abundance of known disease-associated proteins to an anatomical region. HCM = hypertrophic cardiomyopathy; DCM = dilated cardiomyopathy; ARVC = arrhythmogenic right ventricular cardiomyopathy. Proteins highlighted in Box 1 have greater abundance in the intraventricular septum and left ventricle. Box2 proteins have high abundance levels in the right ventricle outer wall. Many of the proteins have low-level abundance in the atrium and valves. (**B**) The 2D heart map images of selected disease-associated proteins showing differing abundance across the heart proteome.

## Data Availability

Data are contained within the article and Appendix A.

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
