# Peer review of "A Proof of Principle 2D Spatial Proteome Mapping Analysis Reveals Distinct Regional Differences in the Cardiac Proteome"

_life, 2024, doi:10.3390/life14080970_

Round 1

Reviewer 1 Report

Comments and Suggestions for Authors

The manuscript "2D Spatial Proteomic Mapping Shows Distinct Regional Differences in the Cardiac Proteome” by Heywood aims to explore the molecular differences, in terms of proteomic profiles” characterizing the different regions of a bovine heart. Although the work and the idea are interesting, as reported by the authors themselves in the Summary section, a limitation concerns the analysis through a single shotgun UDMSe run per sample, limiting the detection of lower abundant proteins. In addition, the analyses were performed on a single biological replicate, and it represents in my opinion a major limitation. 

Major revisions:

-Introduction should be better developed. Similar manuscripts, i.e. Proteomics. 2011 Jun;11(11):2320-8. doi: 10.1002/pmic.201000479, should be cited and evaluated. 

-Methods are poor described. As for UDMSe, the author refers to ref.33. It doesn’t seems a manuscript of the authors, thus they should report in detail the methods used.

-The indication of the PRIDE ID is mandatory.

-The authors state data were acquired in DIA. The don’t mention the spectral library used, but they refer to Bovine Uniprot database. It is not clear how they processed the MS data.

-As for quantitative analysis they mention the use of t-test. However, they are considering more than 2 conditions, thus ANOVA and Tukey test (or other) should be applied. In addition, have they tested if data are normally distributed to apply these tests?

-Figure 3 it is not very readable. Values of variance are not clear.

-Figure 4 reports Abundance. Is it Intensity? PSMs?

-Supplementary data cited in the manuscript are not available. It is not clear how many proteins per sample the authors found.

-Using the profiles characterised, and Protein Atlas, the authors could estimate the proportion of cell types per section. 

-References in the manuscript are not in the right order

-A discussion section should be introduced to discuss the most relevant findings also in relation to the current literature.

-The title and the conclusion of the manuscript should clear highlight that this is an analysis on a single biological sample.

Author Response

Reviewer 1

Comments and Suggestions for Authors

The manuscript "2D Spatial Proteomic Mapping Shows Distinct Regional Differences in the Cardiac Proteome” by Heywood aims to explore the molecular differences, in terms of proteomic profiles” characterizing the different regions of a bovine heart. Although the work and the idea are interesting, as reported by the authors themselves in the Summary section, a limitation concerns the analysis through a single shotgun UDMSe run per sample, limiting the detection of lower abundant proteins. In addition, the analyses were performed on a single biological replicate, and it represents in my opinion a major limitation. 

Major revisions:

-Introduction should be better developed. Similar manuscripts, i.e. Proteomics. 2011 Jun;11(11):2320-8. doi: 10.1002/pmic.201000479, should be cited and evaluated. 

We thaqnk the reviewer for his suggestion and have therefore updated the introduction with more heart proteome literature and re-structured it slightly.

-Methods are poor described. As for UDMSe, the author refers to ref.33. It doesn’t seems a manuscript of the authors, thus they should report in detail the methods used.

As instructed the methods have been updated with further detail. The reviewer maybe be mistaken, the reference 33 is for a citation of the UDMSe technique. The reference Bliss et al (no. 37) has the sufficient detail for the UDMSe method used for this study and is from our group.

-The indication of the PRIDE ID is mandatory.

Apologies the PRIDE information was omitted from the initial manuscript. The PRIDE identifier has now been added and the reviewer account details are as follows;

 Username: [email protected]

 Password: VAjlXQ4T

-The authors state data were acquired in DIA. The don’t mention the spectral library used, but they refer to Bovine Uniprot database. It is not clear how they processed the MS data.

This is a good comment and ‘spot’ from the reviewer and we apologise for this oversight. Data were acquired by MSe DDA not DIA. This has been corrected in the methods section.

-As for quantitative analysis they mention the use of t-test. However, they are considering more than 2 conditions, thus ANOVA and Tukey test (or other) should be applied. In addition, have they tested if data are normally distributed to apply these tests?

The reviewer is correct and we have actually performed the analyses as suggested but didn’t make it clear in the manuscript. The data are processed using Progenesis for proteomics, which performs the data comparisons and normalizes and applies an ANOVA value to proteomics datasets. Cluster comparisons were performed in Progenesis and the ANOVA p value was exported to Excel/Grahpad for creating volcano plots and heatmaps. The manuscript has been updated to clarify this better and we apologise for not making this clearer.

-Figure 3 it is not very readable. Values of variance are not clear.

Figure 3 has been updated with better images

-Figure 4 reports Abundance. Is it Intensity? PSMs?

The protein data itself is intensity data but the term abundance has been used for this figure to show the abundance or amount of the most ‘abundant’ proteins in relation to each other and across the regions. Whilst intensity may be the better technical term abundance gives a lay reader a better impression of the profile of each region. 

-Supplementary data cited in the manuscript are not available. It is not clear how many proteins per sample the authors found.

Apologies the supplementary data has now been provided

-Using the profiles characterised, and Protein Atlas, the authors could estimate the proportion of cell types per section. 

We have identified cell type markers based on the recent single cell paper transcriptomics paper and included our observations in the manuscript and as supplementary figure S4. Koenig, A.L., Shchukina, I., Amrute, J. et al. Single-cell transcriptomics reveals cell-type-specific diversification in human heart failure. Nat Cardiovasc Res 1, 263–280 (2022). https://doi.org/10.1038/s44161-022-00028-6

-References in the manuscript are not in the right order

This has been checked and corrected

-A discussion section should be introduced to discuss the most relevant findings also in relation to the current literature.

The summary has now been edited and integrated into a discussion section

-The title and the conclusion of the manuscript should clear highlight that this is an analysis on a single biological sample.

We have changed the title to ‘A proof of principle 2D Spatial Proteomic Proteome Mapping Analysis Shows Reveals Distinct Regional Differences in the Cardiac Proteome’

Reviewer 2 Report

Comments and Suggestions for Authors

The authors investigated the inter-regional differences of the proteome in a single bovine heart and assessed the feasibility of measuring detailed variance in the cardiac proteome. They identified 3 main distinct proteomes corresponding largely to 1) the outer wall of the right ventricle, 2) the outer wall of the left ventricle, right atrial appendage, tricuspid and mitral valves, modulator band and parts of the left atrium, and 3) the inner walls of the left and right ventricles, septum and left atrial appendage. However, only one heart was used in this pioneering study, and the results are interesting and can provide ideas for further cardiovascular research. The text is easy to follow, and the figures are clear and of good quality. I suggest the acceptance for publication of this MS.

I have only minor formatting issues:

1. The font size is not the same in the MS (e.g., Abstract, lines 31-33; page 6, line 73).

2. There are extra spaces along the text and missing dots at the ends of the sentences.

3. Page 23: information on authors' contribution, conflict of interest, etc., are missing.

Author Response

Reviewer 2

Comments and Suggestions for Authors

The authors investigated the inter-regional differences of the proteome in a single bovine heart and assessed the feasibility of measuring detailed variance in the cardiac proteome. They identified 3 main distinct proteomes corresponding largely to 1) the outer wall of the right ventricle, 2) the outer wall of the left ventricle, right atrial appendage, tricuspid and mitral valves, modulator band and parts of the left atrium, and 3) the inner walls of the left and right ventricles, septum and left atrial appendage. However, only one heart was used in this pioneering study, and the results are interesting and can provide ideas for further cardiovascular research. The text is easy to follow, and the figures are clear and of good quality. I suggest the acceptance for publication of this MS.

I have only minor formatting issues:

  1. The font size is not the same in the MS (e.g., Abstract, lines 31-33; page 6, line 73).

We have checked through the manuscript and corrected any font changes. However, we think some changes may have occurred when the manuscript has been formatted for pdf.

  1. There are extra spaces along the text and missing dots at the ends of the sentences.

This has now been corrected

  1. Page 23: information on authors' contribution, conflict of interest, etc., are missing.

 This has now been included